# Power Control and Clustering-Based Interference Management for UAV-Assisted Networks

**DOI:** 10.3390/s20143864

**Published:** 2020-07-10

**Authors:** Jinxi Zhang, Gang Chuai, Weidong Gao

**Affiliations:** School of Information and Communication Engineering, Beijing University of Posts and Telecommunications, Beijing 100000, China; chuai@bupt.edu.cn (G.C.); gaoweidong@bupt.edu.cn (W.G.)

**Keywords:** UAV communication, coordinate multi-point (CoMP), potential game, affinity propagation

## Abstract

Unmanned Aerial Vehicle (UAV) has been widely used in various applications of wireless network. A system of UAVs has the function of collecting data, offloading traffic for ground Base Stations (BSs) and illuminating coverage holes. However, inter-UAV interference is easily introduced because of the huge number of LoS paths in the air-to-ground channel. In this paper, we propose an interference management framework for UAV-assisted networks, consisting of two main modules: power control and UAV clustering. The power control is executed first to adjust the power levels of UAVs. We model the problem of power control for UAV networks as a non-cooperative game which is proved to be an exact potential game and the Nash equilibrium is reached. Next, to further improve system user rate, coordinated multi-point (CoMP) technique is implemented. The cooperative UAV sets are established to serve users and thus transforming the interfering links into useful links. Affinity propagation is applied to build clusters of UAVs based on the interference strength. Simulation results show that the proposed algorithm integrating power control with CoMP can effectively reduce the interference and improve system sum-rate, compared to Non-CoMP scenario. The law of cluster formation is also obtained where the average cluster size and the number of clusters are affected by inter-UAV distance.

## 1. Introduction

UAV has increasingly become a research hotspot due to its advantages of flexible deployment and low-cost [1]. Leveraging UAVs mounted with antennas as aerial base stations can effectively offload traffic for ground base station and provide seamless coverage for ground users. With the advent of 5G era, the proliferation of mobile devices and the diversification of mobile service will possibly overload ground BSs. Therefore, it is a new and feasible way to relieve the pressure of ground BSs by means of UAV communication. Especially when scenarios like disasters and ground BS failure appear, UAV becomes a reliable choice to provide users with air-to-ground connections [2]. In many types of networks that exist in 5G, such as SWIPT, CRAN and Sensors network, UAV can achieve prominent performance [3].

In the UAV-assisted communication network, UAV can change its location anytime and anywhere according to the distribution density and traffic requirements of users. However, due to the sharp reduction of air obstacles in the high-altitude environment, LoS paths exist widely in air-to-ground channels. Although enhancing the user’s useful signal, it also induces serious interference [4]. Therefore, the most important factor affecting the interference situation in the UAV network is the 3D position of UAVs, including the UAV’s horizontal and vertical coordinates and height. Many studies have also optimized the 3D positions of UAVs to reduce interference and increase network performance. However, changing the position of UAV constantly consumes effective transmission time, resulting in increased transmission delay. In addition, for UAVs with limited battery capacity, this approach is also energy inefficient.

At the same time, CoMP is another key enabler for 5G networks. CoMP builds a coordinated BS set to allow BSs in different locations to cooperate with each other to serve users, thereby suppressing co-channel interference. For edge users, the potential interference is converted into useful signals, hence improving the user performance. Integrating CoMP technology into UAV networks can avoid the strong interference caused by the interfering LoS path, thereby effectively increasing the user transmission rate. This motivates us to design a new CoMP-based architecture for UAV-assisted wireless network.

UAV-assisted network has been extensively investigated in the literature. In [5,6,7,8,9], the UAV deployment was investigated. The authors in [5] formulated the UAV deployment problem from the perspective of coverage probability. The 3D locations of UAVs were optimized to maximize the coverage. In [6], the authors proposed two rapid UAV deployment solutions to minimize the deployment delay. Two cases of UAVs’ starting from the same and different positions were considered to complete the coverage of the target area. The authors in [7] aimed to balance the load of different UAVs and a three-step-based algorithm was implemented. The ground users were first divided into clusters, the centers of which were set to be the initial positions of UAVs. Next, the users associated with the UAVs according to the rate requirements and capacity constraints and finally the 3D locations of UAVs were updated. In [8], the authors also considered the joint optimization of 3D positions and user association. The proposed mechanism first performs UAV-user association while simultaneously considering UAV bandwidth and user-QoS. Furthermore, the 2D positions and heights of UAVs were optimized by k-means and potential-game-based height adjustment, respectively. The authors in [9] decomposed the problem of joint user association and UAV position optimization for a single-UAV scenario into several problems. The non-convexity was solved by replacing it with integer optimization. After that, the positions of the UAVs were optimized by KKT conditions. The authors in [10,11,12] also performed the optimization of trajectory and coverage in various UAV networks.

In addition to location optimization, much research also resorts to the power optimization to alleviate interference and increasing network utility. The authors in [13] studied the joint optimization of 3D locations, user association and power control in a SWIPT system. The complex Mixed-Integer Nonlinear Programming was decomposed into three sub-problems. The UAV-user association and power optimization was respectively solved by binary variable relaxation and successive convex optimization technique. However, the interference issue was not considered in this paper. In [14], the authors maximized the coverage area and energy-efficiency of the UAV network, both of the coverage maximization and power control were proved to be the exact potential game and the Nash Equilibrium was achieved through the formulated utility function. The authors in [15] considered a scenario where UAVs are powered from ground charging station. The joint optimization for sum-rate maximization was realized by iterative optimizing, where variable relaxation and successive convex optimization was also used. The authors in [16] optimized the altitude and power of UAVs in a heterogeneous network. The cross-tier interference between UAVs and other networks including satellite and macrocell was alleviated by optimizing the altitude and power using Lagrange dual decomposition and concave-convex procedure. The authors in [17,18] also improved the network performance of UAV-assited networks by power control and interference management. Moreover, other research issues related to UAV-assisted network have also been widely studied, including energy efficiency [19,20], millimeter-wave [21,22,23] and NOMA [24].

Presently, clustering also become a research hotspot in UAV-assisted networks [25,26,27]. In [25], the authors considered a single-UAV mmwave SWIPT scenario, where two clustering algorithms: K-means and K-medoids are used to perform NOMA to maximize the harvested power for ground users. In [26], affinity propagation was applied to cluster users, the selected exemplars are required to lower their transmission power to increase network energy-efficiency. K-means was also used to adjust the positions of UAVs to further increase the QoS of users. In [27], ground users were divided into clusters, each of which was iteratively served by UAVs through cooperative transmission. Through deriving the upper and lower bound of user rate by approximation calculation and applying KKT conditions, the 2D positions of UAVs under static and dynamic deployment cases were both optimized. However, inter-cluster interference was neglected and clusters were served in an alternative manner and therefore degrading the network utility.

Although previous works have demonstrated the huge effect of UAV-BSs on network performance, there still exist challenges to be studied and solved. To be concrete, UAVs continuously change their locations which lead to the energy-inefficiency and increased transmission delay, hence the interference management under fixed locations is urgently needed. In addition, the nature of mitigating interference and strengthening transmission quality in CoMP [28,29,30] has not been fully applied in UAV systems. The regularity of cluster forming in UAV networks has not been thoroughly studied, including average cluster number and average cluster size (the number of UAVs in a cluster). The two metrics can effectively reveal the relationship between cluster formation and network topology, interference issue and user distribution. In fact, there exists little research regarding CoMP-based UAV system and the benefit of CoMP combining UAV network is not fully reaped.

Motivated by previous works, we design a CoMP-based power optimization and clustering algorithm in UAV-assisted network. We first propose a power control algorithm which is proved to be an exact potential game. The utility function is designed as the summation of the rates of player’s neighbors. Next, Affinity Propagation-based clustering algorithm is leveraged to divide UAVs into clusters. The users in the same cluster share the transmission from a same group of UAV-BSs leveraging zero-forcing precoding algorithm. The main contributions of this paper are summarized as follows:We propose an interference management framework in UAV cooperative transmission architecture for UAV-assisted network. UAVs can jointly transmit desired signals for the ground users in the cluster via zero-forcing beamforming technique. The problem is formulated as the joint optimization of power control and UAV clustering. A two-step-based optimization is proposed to optimize the power allocation and UAV grouping process.We propose a potential game-based power control(PC) algorithm to optimize the power allocation of the UAVs. The power control problem is proved to be an exact potential game and the Nash Equilibrium is reached. At first, by fixing a RSRP threshold, the rule of determining neighbors is defined as follows [8]: if there exists at least a ground user whose RSRP strength from two UAVs is simultaneously stronger than the given RSRP value, then the two UAVs are regarded as neighbors. After that, the power strategies are continuously updated until the convergence is reached, and the final optimal power allocation result is  obtained.We propose an affinity propagation-assisted UAV clustering algorithm by means of Affinity propagation clustering (APC). UAVs and their associated users are clustered into groups according to the mutual interference and the UAVs who potentially receive severe interference from each other have a stronger tendency to join the same cluster. In the initial stage of APC, matrixes of similarity, availability and responsibility is constructed. During the APC algorithm, three matrixes are update continuously until the convergence criterion is reached. Finally, UAVs are designed to select their cluster head according to the final matrixes, and thus the optimal clusters are formed. ZF precoding is used to mitigate intra-cluster interference and improve user rate.Numerical results are carried out in static-UAV scenarios to validate the effectiveness of our proposed algorithm. Under different parameters, sum-rate result shows that the proposed algorithm can prominently increase network sum-rate compared with non-CoMP case. Moreover, our results also reveal the law of cluster formation including the average number of clusters and average cluster size under the case with or without minimum inter-UAV distance constraints. In a word, our algorithm can also effectively avoid strong interference and increase user rate by power control and clustering.

The rest of this paper is organized as follows: In Section 2, we introduce the system model and formulate the problem of joint optimization. In Section 3, we give the detailed process of our proposed interference management framework. In Section 4, we illustrate our proposed two-step algorithm by applying potential game theory and affinity propagation. In Section 5, the numerical results are discussed. At last, we draw the conclusion in Section 6.

## 2. System Model

As shown in Figure 1, we consider a UAV-enabled network in this paper, where *M* rotary-wing UAVs are deployed in the target area. The set of UAVs is denoted as M={1,2,…,M}, and the set of ground users is denoted as K={1,2,…,K}. Each UAV is equipped with *N* element uniform linear array (ULA) and each ground user is equipped with a single antenna. We focus on the downlink transmission scenario, where UAVs form clusters to serve a group of users through cooperative transmission. We assume that UAVs exchange the information with Central Unit (CU) through wireless backhaul links and hence the instantaneous channel state information(CSI) between UAVs and users is available. In addition, under the scheduling of CU, ZF precoding is adopted at UAVs for signal transmission. The 3D Cartesian coordinates of UAV *m* and ground user *k* is denoted by $\left[x_{m},y_{m},h_{m}\right]$ and $\left[x_{k},y_{k},0\right]$ respectively. The height of user is fixed at 0 and the height of UAV *m* follows uniform distribution: $h_{m}∼N\left(50,150\right)\;m$. It is worthwhile to note that we consider a static-UAV scenario, where the locations of UAVs and ground users are static and not varied during the execution of algorithm.

The distance of UAV *m* and user *k* is expressed as:(1)$$d_{m,k}=\sqrt{{\left(x_{m}-x_{k}\right)}^{2}+{\left(y_{m}-y_{k}\right)}^{2}+h_{m}^{2}}$$

As stated before, LoS links widely exist in the UAV-UE channels. However, there is still possibility for air-to-ground links to be NLoS due to obstacles. Intuitively, the LoS or NLoS probability between UAVs and ground users is determined by the environment and the relative positions. The probability of LoS link between UAV *m* and user *k* is calculated as:(2)$$P_{LoS}=\frac{1}{1+aexp(-b[\epsilon -a])}$$
where ϵ is the elevation angle between UAV *m* and UE *k*, and *a* and *b* are environment-related constants. Therefore, the channel gain between UAV *m* and UE *k* is denoted as:(3)$$PL_{m,k}=P_{LoS}d_{m,k}^{-\alpha }+\beta P_{LoS}d_{m,k}^{-\alpha }$$
where α is the pathloss exponent, $P_{NLoS}=1-P_{LoS}$ is the probability for having a NLoS path and β is the additional attenuation factor for NLoS links. The channel gain vector between UAV *m* and user *k* is denoted by $h_{m,k}\in C^{1\times N}$:(4)$$h_{m,k}=PL_{m,k}{\left[1,e^{j\pi \theta_{m,k}},\ldots ,e^{j\left(N-1\right)\pi \theta_{m,k}}\right]}^{T}$$
where $\theta_{m,k}$ is the AoD (Angle of Departure) of link m−k, and $\theta_{m,k}$ follows U(−1,1). The channel matrix from UE *k* to all the UAVs is $H_{k}\in C^{M\times N}={\left[h_{1,k}^{T},h_{2,k}^{T},\ldots ,h_{M,k}^{T}\right]}^{T}$, thus the precoding matrix for UE *k* after ZF precoding algorithm is denoted as:(5)$$V_{k}=H_{k}^{†}=H_{k}^{*}{\left[H_{k}H_{k}^{*}\right]}^{-1}$$
where $V_{k}\in C^{N\times M}=\left[v_{k,1},v_{k,2},\ldots ,v_{k,M}\right]$ and $v_{k,m}$ is the m−th column of $V_{k}$ denoting the beamforming vector for link m−k.

After performing the clustering algorithm, we assume the network is divided into *C* clusters, and the set of clusters is denoted as: $C=\{\Phi_{1},\Phi_{2},\cdots ,\Phi_{C}\}$. Therefore, the received signal at user *k* who is connected to cluster $\Phi_{k}$ is denoted as:(6)$$\begin{array}{cc} y_{k} & =\sum_{\Phi_{c}\in C}\sum_{m\in B_{\Phi_{c}}}\sum_{k\in U_{\Phi_{c}}}p_{m}h_{m,k}v_{k,m}\\ \; & =\sum_{m\in B_{\Phi_{k}}}p_{m}h_{m,k}v_{k,m}+\sum_{\Phi_{k^{\prime }}\neq \Phi_{k}}\sum_{m^{\prime }\in B_{\Phi_{k^{\prime }}}}\sum_{j\in U_{\Phi_{k^{\prime }}}}p_{m^{\prime }}h_{m^{\prime },k}v_{j,m^{\prime }}+n_{0}B \end{array}$$
where $p_{m}$ is the transmission power of UAV *m*. $B_{\Phi_{k}}$ and $U_{\Phi_{k}}$ are the BS set and the user set of cluster $\Phi_{k}$, respectively. The first part in (Equation 6) is the useful signal transmitted by user *k*’s cooperative UAV set, the second part is the inter-cluster interference. $n_{0}$ is the noise power and *B* is the system bandwidth.

Hence, the SINR for user *k* is calculated as:(7)$$\begin{array}{c} \gamma_{k}=\frac{\sum_{m\in B_{\Phi_{k}}}p_{m}h_{m,k}v_{k,m}}{\sum_{\Phi_{k^{\prime }}\neq \Phi_{k}}\sum_{m^{\prime }\in B_{\Phi_{k^{\prime }}}}\sum_{j\in U_{\phi_{k^{\prime }}}}p_{m^{\prime }}h_{m^{\prime },k}v_{k^{\prime },m^{\prime }}+n_{0}B} \end{array}$$

## 3. Problem Formulation

Next, we formulate the problem of joint optimization of power control and UAV clustering. The target function is designed as the maximization of the sum-rate of all the users in the network, subject to the feasible power range and the cluster formation restrictions.
(8)$$\begin{array}{cc} \underset{p_{m},C}{maximize} & \sum_{k\in K}\log_{2}\left(1+\gamma_{k}\right)\\ s.t.\;\;C1 & :p_{min}\leq p_{m}\leq p_{max},\forall m\in M\\ C2 & :\Phi_{i}\cap \Phi_{j}=⌀,\forall i\neq j\\ C3 & :\cup_{\Phi_{i}\in C}B_{\Phi_{i}}=M \end{array}$$
where C1 indicates that the power of each UAV should falls into the reasonable range. C2 indicates that there is no overlap between different clusters and C3 indicates that the final cluster sets should contain all the UAVs.

It can be seen from (Equation 7) that the user rate is mainly influenced by the power of UAVs, the channel gains form the UAVS and the clustering result. For a ground user, the interference comes from the UAVs outside the its cooperative cluster, because of the cancellation of intra-cluster. In the next section, we illustrate how to reduce the interference and boost the overall user rate in the network by power control and UAV clustering.

## 4. Interference Management Framework for UAV-Assisted Network

In this section, our designed interference management framework is proposed. To reduce the interference and improve user rate, we use potential game-based power control to adjust the power level of UAVs. With the help of CU located at ground, the interaction between UAVs and the whole process of the algorithm including PC and APC is implemented. Specifically, CU functions as the collector of network data, the executor of the algorithm and the deliverer of parameter adjustment information. The interference management framework mainly includes the following steps: data collection, data cleaning, power control for UAVS, Affinity-propagation clustering for UAVs and parameter reconfiguration.

The detailed process of the framework is presented in Figure 2 and each step is explained as follows:Data collection: CU collects the network operation data, which includes: UAV ID, UE ID, RSRP information, the power levels of UAVs, etc.Data cleaning: The useful information is extracted from the collected data and the extra information that cannot be used for interference control is discarded. Next, the useful data is arranged in a requested format to facilitate the operation of next step.Power control: Based on the arranged data, CU first finds neighbors for each UAV and establish the potential game model. The power levels of UAVs are optimized to maximize the network utility. After the convergence of the potential game, the final power control results are delivered to the next step.Affinity Propagation clustering: In this stage, CU performs UAV clustering using Affinity Propagation to reduce interference. AP is an algorithm that can automatically determine the cluster number and cluster heads based on the interference relationships. The similarity matrix is calculated first and availability and responsibility are iteratively updated. Based on the converged matrixes, the cluster heads and cluster members are selected, thus forming the clusters.Parameters reconfiguration: The final power levels and clustering results are delivered to the UAVs. At UAV side, power adjustment and cooperative transmission are executed according to the final clustering  result.

## 5. Proposed Algorithm

In this section, we illustrate the whole process of our proposed algorithm including potential game-based power control and Affinity-Propagation-based clustering. Before the execution of the algorithm, we illustrate the process of UAV-UE association. Then the potential-game-based power control algorithm is proposed to adjust the power lever of UAVs to alleviate the interference. Lastly, the APC algorithm is used to form clusters of UAVs to build cooperative BS set for users to mitigate intra-cluster interference.

### 5.1. UAV-UE Association

At the initial stage of our algorithm, each UE first choose the UAV with the best channel condition after the UAV positions are fixed. In other words, UE *m* choose the UAV that can provide the maximum RSRP for UE *m*.
(9)$$\Phi_{k}=\underset{m\in M}{argmax}\;p_{m}\;{\left|h_{m,k}\right|}^{2}$$

Please note that at the association stage, the clusters are not formed yet. UE *k* associates to only one UAV and hence the cooperative set $\Phi_{k}$ only contains a single UAV.

### 5.2. Potential Game-Based Power Control

After the UAV-UE association, we perform the power control to decrease the interference and improve network utility. Obviously, the change of the power of a single UAV does not only impact its users’ rate but also cause impact on the users severed by other UAVs. Hence, the optimal power strategy of the whole network is derived from the power control of all the UAVs. Potential game has the natural advantage of multi-agent decision making and can obtain the optimal global utility by local interaction. In our Potential Game model, UAVs modeled as players keep changing their power strategies until the Nash Equilibrium is achieved. The utility function is designed that the maximization of user utility corresponds to the maximization of network utility. Next, we will illustrate in detail the model, utility function and the whole process of the algorithm.

The problem of power control in UAV-assisted networks is formulated as the game $G=\{M,{\left\{A_{m}\right\}}_{m\in M},{\left\{u_{m}\right\}}_{m\in M}\}$, where M is the set of players, i.e., the UAV set. $A_{m}$ is the power set of feasible actions of player *m*, and $u_{m}$ is the utility function of player *m*. For the reason that our optimized variables are the power levels of the UAVs, $A_{m}$ is regarded as the set of feasible power value for the UAVs.

Next, we define the neighbors for each player. For two UAVs *m* and $m^{\prime }$, if there exist at least one UE *k*, that satisfy:(10)$$p_{m}\;{\left|h_{m,k}\right|}^{2}>TH_{RSRP}\;\&\;p_{m^{\prime }}\;{\left|h_{m^{\prime },k}\right|}^{2}>TH_{RSRP}$$
then *m* and *l* are regarded as neighbors. Therefore, the neighbor set of players *m*-$N_{m}$ is denoted as:(11)$$\begin{array}{cc} N_{m} & =\{l\in M,l\neq m|\exists k\in K,\\ s.t.\; & p_{m}\;{\left|h_{m,k}\right|}^{2}>TH_{RSRP}\;\&\;p_{l}\;{\left|h_{l,k}\right|}^{2}>TH_{RSRP}\} \end{array}$$

The utility function for player *m* is designed as:$$\begin{array}{cc} U_{m}\left(P\right) & =\sum_{k\in U_{m}}\log_{2}\left(1+\bar{\gamma_{k}}\right)+\sum_{l\in N_{m}}\sum_{k\in U_{l}}\log_{2}\left(1+\bar{\gamma_{k}}\right)\\ \; & =\sum_{k\in U_{m}}\log_{2}\left(1+\frac{p_{m}h_{m,k}v_{k,m}}{\sum_{m^{\prime }\in N_{m}}\sum_{j\in U_{m^{\prime }}}p_{m^{\prime }}h_{m^{\prime },k}v_{j,m^{\prime }}+n_{0}}\right)\\ \; & +\sum_{l\in N_{m}}\sum_{k\in U_{l}}\log_{2}\left(1+\frac{p_{l}h_{l,k}v_{k,l}}{\sum_{l^{\prime }\in N_{l}}\sum_{j\in U_{l^{\prime }}}P_{l^{\prime }}h_{l^{\prime },k}v_{j,l^{\prime }}+n_{0}}\right) \end{array}$$
where $P=\{p_{1},p_{2},\cdots ,p_{M}\}$ is the power vector of the UAVs. As can be seen, $\bar{\gamma_{k}}$ is the approximated SINR for user $k\in U_{m}$ when only the interference from its neighbor BSs is considered. Then we give the definition of the potential function, which is modeled as the summation of approximate rate of all the users in the network:(12)$$\begin{array}{c} F\left(P\right)=\sum_{m\in M}f_{m}=\sum_{m\in M}\sum_{k\in U_{m}}\log_{2}\left(1+\bar{\gamma_{k}}\right) \end{array}$$
where $f_{m}=\sum_{k\in U_{m}}\log_{2}\left(1+\bar{\gamma_{k}}\right)$ is the approximate sum-rate of all the users in UAV *m*.

**Definition** **1**(Nash Equilibrium [31]). *A power strategy profile $P^{*}=\left(p_{1}^{*},p_{2}^{*},\ldots ,p_{M}^{*}\right)$ is a Nash equilibrium point of G if and only if no player can improve its utility by changing its power strategy unilaterally. That is for $\forall m\in M,\forall p_{m}\in A_{n},p_{m}\neq p_{m}^{*}$, we have:*
(13)$$\begin{array}{c} u_{m}\left(p_{m}^{*},p_{-m}^{*}\right)>u_{m}\left(p_{m},p_{-m}^{*}\right) \end{array}$$
*where $p_{-m}$ denotes the power strategy of all the UAVs excluding UAV m.*


**Definition** **2**(Exact Potential Game [32]). *If the potential function F(P) of game G satisfies that: for each player m, $\forall (p_{m},p_{-m})$ and $\left(p_{m}^{\prime },p_{-m}\right)$:*
(14)$$\begin{array}{c} F\left(p_{m},p_{-m}\right)-F\left(p_{m}^{\prime },p_{-m}\right)=u_{m}\left(p_{m},p_{-m}\right)-u_{m}\left(p_{m}^{\prime },p_{-m}\right) \end{array}$$
*then game G is an exact potential Game. Equation (Equation 14) indicates that for potential game-based model, the increase of user utility will also lead to the increase of network utility. Hence, continue improving the user utility will finally bring the game to the NE point.*


**Proposition** **1.**
*Our proposed game G is an exact potential game and the function F defined by Equation (Equation 12) is a potential function.*


**Proof.** After changing the power strategy $p_{m}$ of an arbitrary player *m*, the function value changes, and the change value can be decomposed as follows:
(15)$$\begin{array}{cc} \; & F\left(p_{m},p_{-m}\right)-F\left(p_{m}^{\prime },p_{-m}\right)\\ \; & =\sum_{m\in M}\left(f_{m}\left(p_{m},p_{-m}\right)-f_{m}\left(p_{m}^{\prime },p_{-m}\right)\right)\\ \; & =f_{m}\left(p_{m},p_{-m}\right)-f_{m}\left(p_{m}^{\prime },p_{-m}\right)\\ \; & +\sum_{j\in N_{m}}f_{j}\left(p_{m},p_{-m}\right)-f_{m}\left(p_{m}^{\prime },p_{-m}\right)\\ \; & +\sum_{j\notin N_{m},j\neq m}f_{j}\left(p_{m},p_{-m}\right)-f_{m}\left(p_{m}^{\prime },p_{-m}\right) \end{array}$$Obviously, the power adjustment of player *m* will not impact the utility of the UAVs that does not belong to the neighbor set of *m*, and hence $\sum_{j\notin N_{m},j\neq m}f_{j}\left(p_{m},p_{-m}\right)-f_{m}\left(p_{m}^{\prime },p_{-m}\right)=0$. Therefore, we have:
(16)$$\begin{array}{cc} \; & F\left(p_{m},p_{-m}\right)-F\left(p_{m}^{\prime },p_{-m}\right)\\ \; & =f_{m}\left(p_{m},p_{-m}\right)-f_{m}\left(p_{m}^{\prime },p_{-m}\right)\\ \; & +\sum_{j\in N_{m}}f_{j}\left(p_{m},p_{-m}\right)-f_{m}\left(p_{m}^{\prime },p_{-m}\right)\\ \; & =u_{m}\left(p_{m},p_{-m}\right)-u_{m}\left(p_{m}^{\prime },p_{-m}\right) \end{array}$$By far, we can finalize the proof that game G is an exact potential game and has at least one pure Nash Equilibrium point. □

Algorithm 1 is the detailed process of power control. During the process of potential game, the players continuously explore the feasible power strategy set and choose the exploration action $p_{m}^{e}$ or keep their previous power strategy $p_{m}^{o}$ based on the utility. For UAV *m*, while other UAVs do not change their strategies, the possibility of choosing a new exploration action $P_{m}^{e}$ is formulated as:(17)$$\begin{array}{c} Prob_{m,e}=\frac{exp(\varphi F\left(p_{m}^{e},p_{-m}\right))}{exp\left(\varphi F\left(p_{m}^{e},p_{-m}\right)\right)+exp\left(\varphi F\left(p_{m}^{o},p_{-m}\right)\right)} \end{array}$$
where φ is the exploration factor. In other words, UAV *m* will keep its previous power strategy with possibility $1-Prob_{m,e}$.
**Algorithm 1** Potential game-based power control**Input:** players M, feasible power set $P_{f}$, maximum iteration number ITER**Output:** optimal power strategy $P_{opt}$1:Set current iteration iter=0.2:**while**iter < ITER
**do**:3:    Randomly choose a player m∈M to change its power strategy.4:    Record its power strategy of previous iteration as $p_{m}^{o}$.5:    Choose an exploration action $p_{m}^{e}$ from its action set.6:    Calculate the potential utility under $p_{m}^{e}$ and $p_{m}^{o}$.7:    UAV *m* update its power strategy according to (Equation 17).8:    iter = iter + 1;9:**end while**10:**return** The final power strategy of all the UAVs $P_{opt}=\left\{p_{1}^{*},p_{2}^{*},\ldots ,p_{M}^{*}\right\}$.

### 5.3. Affinity-Propagation Based Clustering Algorithm for UAVs

In this section, we apply APC-based CoMP technique in UAV-assisted networks. The interference issue is much improved after previous power control stage; however, edge users located at the brink of the UAV coverage still have a high possibility to suffer severe interference from near-distant neighbor UAVs. Hence, after optimizing the power of all the UAVs, we perform UAV clustering to divide UAVs into several clusters. For users in the cluster especially edge users, the interference from neighbor UAVs are eliminated and the user rate is further  improved.

Next, we introduce the concept of cluster members and cluster exemplars. In the UAV-assisted network, if there exist severe interference between two UAVs, they have a strong tendency to be in the same cluster. Compared with cluster members, cluster exemplar suffers the strongest interference and shows the strongest willingness to form clusters with its neighbor UAVs. However, how to appropriately cluster UAVs, find the exemplars heads and reduce the interference to the largest extent remains a challenge. In this context, APC provides an unsupervised learning technique to automatically determine the number of clusters and the cluster exemplars.

First, the similarities of UAVs are calculated to characterize the interference relationship. In a network consisting of *M* UAVs, the similarity matrix *S* is denoted as:(18)$$\left(\begin{array}{ccccc} s_{1,1} & \cdots & s_{1,l} & \cdots & s_{1,M}\\ \vdots & \ddots & \vdots & \ddots & \vdots \\ s_{m,1} & \cdots & s_{m,l} & \cdots & s_{m,M}\\ \vdots & \ddots & \vdots & \ddots & \vdots \\ s_{M,1} & \cdots & s_{M,l} & \cdots & s_{M,M} \end{array}\right)$$
where *m* and *l* are the rows and columns of *S*. The value $s_{m,l}$ is calculated as RSRP summation of the users in UAV *m* received from UAV *l*, characterizing the degree of interference that UAV *l* impose on the users of *m*:(19)$$\begin{array}{c} s_{m,l}=\sum_{k\in U_{m}}p_{l}{\left|h_{l,k}\right|}^{2} \end{array}$$

At the beginning of the algorithm, the elements of similarity matrix *S* is calculated. As shown in Figure 3, the process of responsibility and availability are updated iteratively. The $responsibility\;r_{m,l}$ is sent from UAV *m* to UAV *l*. Obviously, a bigger value of $r_{m,l}$ indicates that UAV *l* is more suitable to be the exemplar of *m*. $r_{m,l}$ reflects the accumulated evidence of how well-suited for UAV *l* to be the exemplar of UAV *m*, taking other candidate exemplars into consideration. $responsibility\;r_{m,l}$ is calculated as the following rule:(20)$$\begin{array}{c} r_{m,l}=s_{m,l}-max_{l^{\prime }\neq l}\left\{a\left(m,l^{\prime }\right)+s\left(m,l^{\prime }\right)\right\} \end{array}$$
where $a(m,l^{\prime })$ is the availability sent from candidate exemplar $l^{\prime }$ to UAV *m*. a(m,l) reflects the accumulated evidence of how well-suited for UAV *m* to choose *l* as its exemplar. The availability is updated as follows:(21)$$\begin{array}{c} a_{m,l}={\left(r_{l,l}+\sum_{m^{\prime }\in M,m^{\prime }\notin m,l}max\left\{0,r\left(m^{\prime },l\right)\right\},0\right)}^{+} \end{array}$$
where ${\left[x\right]}^{+}=max\left\{x,0\right\}$.

The execution process of APC is illustrated as follows: first, the matrix *S* is calculated as (Equation 19), *R* and *A* are initialized as zero matrixes. Next, the responsibility and availability is updated. Finally, the criterion matrix is established as the summation of responsibility and availability matrixes. For each UAV *m*, the UAV with the maximum criterion value in column *m* is selected as the exemplar. The detailed process can be seen in Algorithm 2.
**Algorithm 2** Affinity-Propagation clustering for UAV-assisted network**Input:** The set of UAVs M={1,2,⋯,M}, the set of UEs K={1,2,⋯,K}, maximum iteration number ITER, current iteration index iter**Output:** The set of clusters: $\Phi =\{\Phi_{1},\Phi_{2},\cdots ,\Phi_{c}\}$1:Calculate similarity matrix $S={\left[s_{m,l}\right]}_{M\times M}$ according to (Equation 19), and initialize $responsibility\;R={\left[0\right]}_{M\times M}$ and $availability\;A={\left[0\right]}_{M\times M}$.2:**repeat**3:    Update $R=\{r_{m,l}\}$ according to (Equation 20) and broadcast.4:    Update $A=\{a_{m,l}\}$ according to (Equation 21) and broadcast.5:    iter=iter+1.6:**until** convergence or iter=ITER.7:**for** each UAV l∈M
**do**8:    $exemplar\left(l\right)=argmax_{m\in M}\left\{a_{m,l}+r_{m,l}\right\}$9:**end for**

## 6. Numerical Results

In this section, we evaluate the performance of our proposed algorithm. The simulation parameters are set as Table 1. The target area is set to the circle with radius of 500m and all the UAVs and ground users are distributed in the target area. In our experiment, we generate multiple snapshots and UAVs and UEs are regaded as stationary points in each snapshot. We also give the regularity of cluster formation, including average cluster size and the number of clusters by varying the number of UAVs and UEs. The performance of user rate is also evaluated by comparing the CDF curve and system sum-rate for CoMP and Non-CoMP scenarios.

The final power of UAVs are shown in Figure 3. It can be seen that the power of all the UAVs have been adjusted to increase network utility. Initially, the power is set to 38 dBm for all the UAVs. During the process of potential game, the power of UAVs are iteratively updated in the direction of increasing network utility. Finally, the convergence of game finalize the power level of UAVs.

In Figure 4, we draw a typical clustering result of the network under two cases with and without minimum distance restriction $d_{min}$, fixing M=10,K=50. It shows that after APC, the UAVs that cause interference to each other are more likely to join the same cluster to reduce interference, which is highly related to the inter-UAV distance. A smaller distance will result in a bigger probability of forming a cluster to avoid interference. Figure 4 also indicates that the scenario without $d_{min}$ will potentially generate more clusters compared to the scenario with $d_{min}$. Under no distance restriction, the UAVs formed 3 clusters each contains UAV of 3,2,2, respectively. In contraction, under $d_{min}=150\;m$, only 2 clusters are formed, each contains 2 UAVs. The reason lies in that the average distance between UAVs is shortened, the interference issue is more severe and hence the possibility of forming clusters is increased.

To further illustrate the regularity of cluster size and the number of clusters, we perform 50 experiments in the same target area, which generate 50 snapshots of network to simulate the diverse distribution of UAVs and users, and obtain the average performance. In Figure 5, we show $Avg_{size}$ (the average cluster size of 50 snapshots) and $Avg_{num}$ (the average number of clusters whose size>1 in each snapshot) under two cases. First, we fix the number of UAVs and ground users (M,K) to (10,50), respectively. The $Avg_{size}$ and $Avg_{num}$ for case $d_{min}=0$ is 1.315 and 2.2 while those for $d_{min}=150$ is 1.190 and 1.46. This indicates that the close-distance results in severe interference and increase the tendency for UAVs to join the same cluster, thus boosting the number of clusters and cluster size. Furthermore, we put more UAVs and users into the network. When M,K increases from (10,50) to (15,80), $Avg_{size}$ is increased from 1.315 to 1.612 for $d_{min}=0$ and from 1.190 to 1.209 when $d_{min}$ exist. Similarly, $Avg_{num}$ also steps up for both cases with and without $d_{min}$. As more UAVs and users are involved in the network, the inter-UAV distance further decreases and the severe interference is incurred. It can be observed that the existence of $d_{min}$ can alleviate interference and reduce cluster size and the number of clusters. Therefore, more clusters are formed, and the average cluster size grows with the number of UAVs and users. In addition, we also show the standard deviation (SD) of cluster size ($SD_{size}$) and the number of clusters ($SD_{num}$). It can be seen that under $M=15,d_{min}=0$, the network has the biggest SD of cluster size. This is because the number of clusters grows and the cluster size are various, hence the deviation of cluster size are increased. However, under $M=15,d_{min}=0$, the cluster size is 1 in most cases, which results a small deviation. Furthermore, as the number of clusters (whose size>1) is usually 2, 3, and very few are 1 in each snapshot, the standard deviation of the number of clusters is very small for all the scenarios.

Next, we present the Cumulative Distribution Function (CDF) of user rates (Mbit/s) in Figure 6. We compare the CDF curves of three cases: non-CoMP, Power control only and Power control + CoMP. The figure indicates that non-CoMP shows the worst performance compared to other two algorithms. The reason lies in that there are a huge number of interfering UAVs and UEs, and these interfering links will degrade the user performance. After potential-game-based power control, the power levels of UAVs are adjusted, and the interference is reduced. Hence, the user rate performance under PC is much improved compared with non-CoMP. Among the three algorithms, PC + CoMP shows the best performance. Based on power control, the APC algorithm can further reduce interference by forming cooperative UAV sets. The cooperative transmission transforms the interfering links into useful links, and the interference in the same cluster is mitigated.

Finally, the impact of the number of users on network performance is illustrated in Figure 7. The number of UAVs is fixed to 10, and we increase the number of ground users to evaluate its performance on system sum-rate. When *K* increases from 20 to 50, the system sum-rate decreases from 35.26 to 20.85 under CoMP, while the system sum-rate decreases from 32.17 to 10.97 for Non-CoMP case. The proliferation of users in the network make the interference more complicated. Although the intra-cluster interference can be eliminated by ZF precoding, the inter-cluster interference grows stronger due to the increase of *K*. However, it can also be observed from the figure that the system sum-rate under CoMP still outperforms that of non-CoMP. Therefore, we can conclude that the combination of power control and CoMP shows superiority of alleviating interference and enhancing network performance.

## 7. Conclusions

In this paper, we introduce our proposed interference management framework integrated with power control and UAV clustering. The target problem is formulated as the maximization of system sum-rate. The power levels of UAVs are adjusted to reduce interference and affinity-propagation-based clustering algorithm is further implemented to further improve user rate by mitigating intra-cluster interference. Numerical results validate the effectiveness of the algorithm. The system sum-rate are prominently enhanced compared with Non-CoMP scenario. The system performance still maintains at a good level when the UEs in the network quickly grows, which show the superiority of CoMP. The results also reveal the regularity of average cluster size and the number of clusters and the minimum distance restriction is proved to have the function of reducing interference and preventing the formation of cluster. CoMP-based UAV system has the advantage of mitigating interference and promote network performance. In the future research, the combination of UAV and CoMP technique can be applied in more and more scenarios.

## Figures and Tables

**Figure 1 sensors-20-03864-f001:**
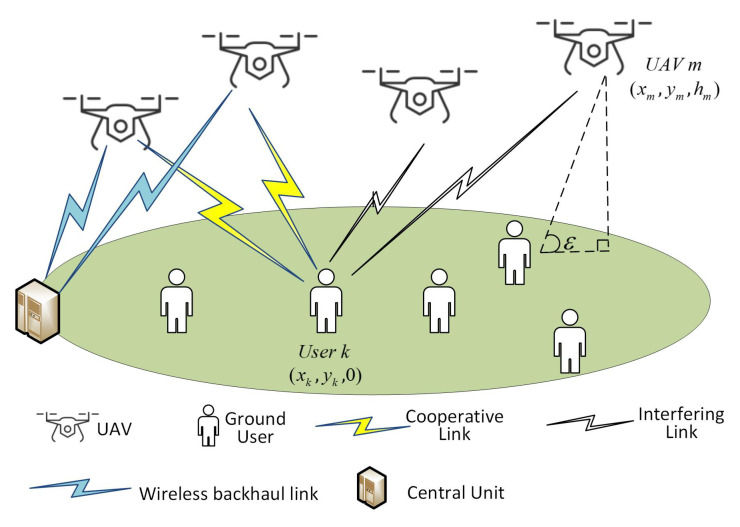
System Model.

**Figure 2 sensors-20-03864-f002:**
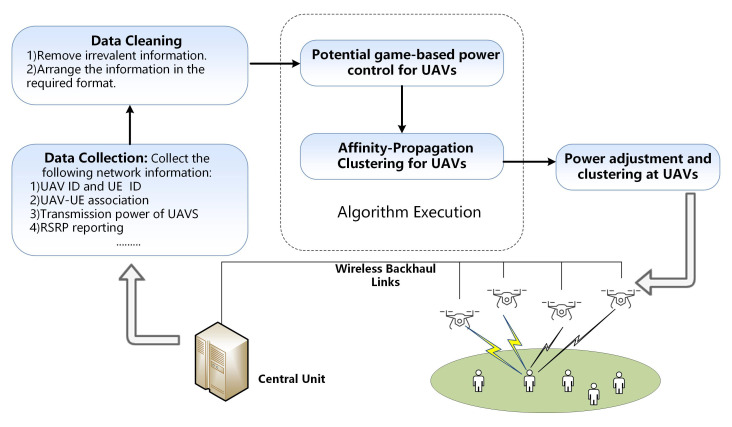
Interference management process.

**Figure 3 sensors-20-03864-f003:**
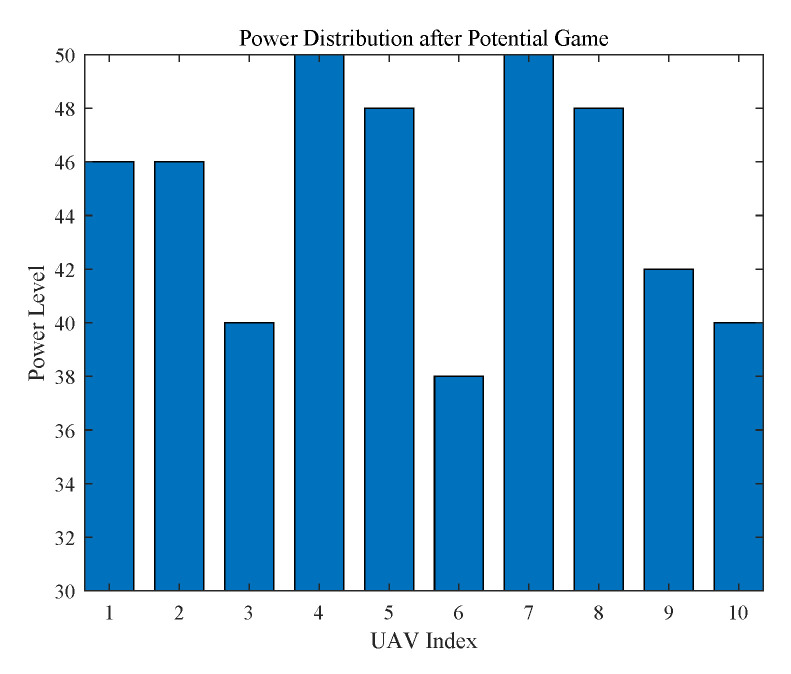
Final Power distribution after PG.

**Figure 4 sensors-20-03864-f004:**
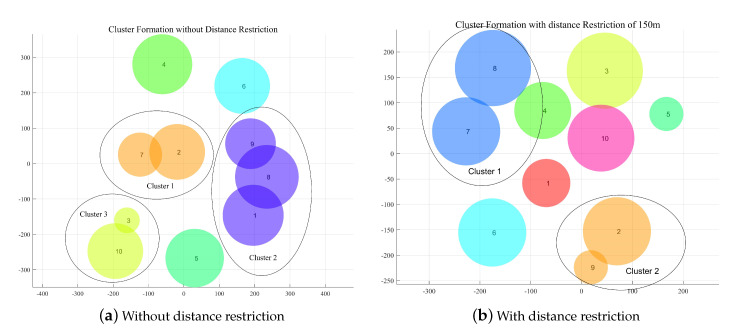
Comparison of cluster result for two cases: with and without minimum distance restriction. (**a**) Without distance restriction; (**b**) With distance restriction.

**Figure 5 sensors-20-03864-f005:**
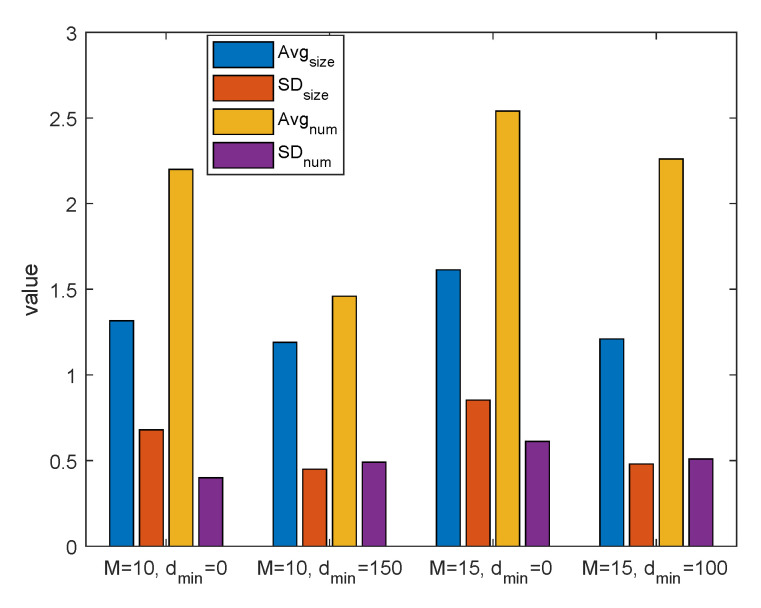
Average cluster size and the number of clusters.

**Figure 6 sensors-20-03864-f006:**
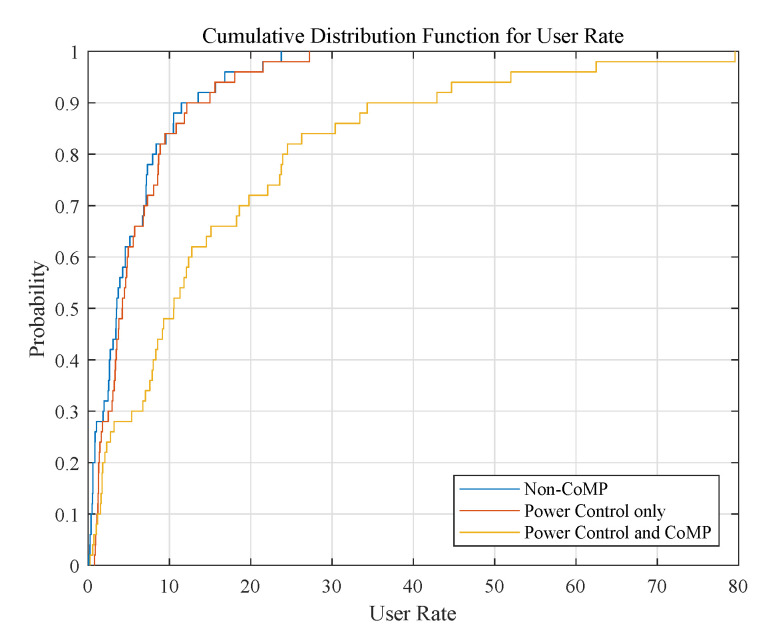
CDF plot for three cases: Non-CoMP, Power control only and Power control + CoMP.

**Figure 7 sensors-20-03864-f007:**
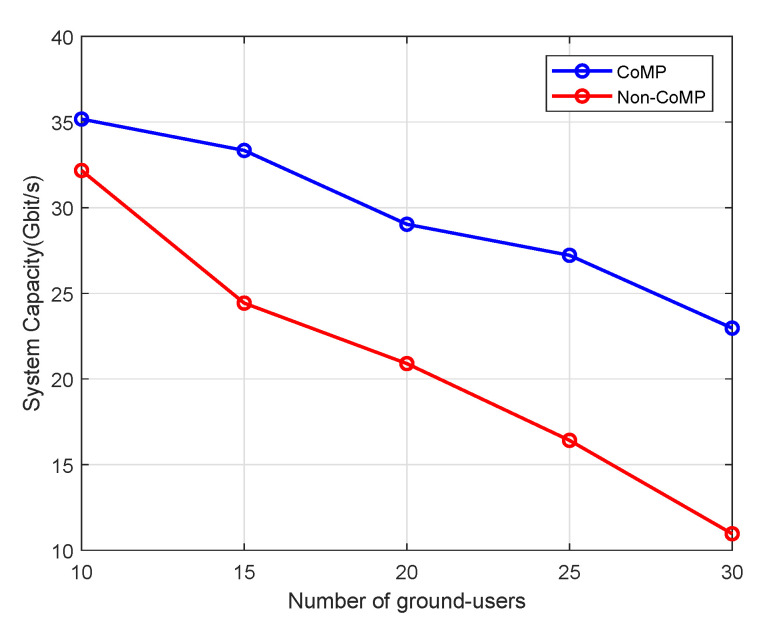
Sum-rate comparison of CoMP and Non-CoMP under different number of users K.

**Table 1 sensors-20-03864-t001:** Simulation Parameters.

Parameter	Value
Area radius (m)	500
System bandwidth (MHz)	10
Pathloss exponent, α	2
Attenuation factor, β	0.01
Exploration factor, φ	10
Power set, $P_{f}$ in dBm	(38,40,42,44,46,48,50)
RSRP threshold, $TH_{RSRP}$ (dBm)	−85
Noise power, $n_{0}$ (dBm/Hz)	−174
Maximum iteration, ITER	1000
Channel parameters, a,b	11.95, 0.136

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
