# Peer review of "Power Control and Clustering-Based Interference Management for UAV-Assisted Networks"

_sensors, 2020, doi:10.3390/s20143864_

Round 1

Reviewer 1 Report

Dear Authors,

this paper deals with algorithms to manage a network of UAV that cooperate to trasmit/receive data to ground nodes. Such algorithm is based on a non-cooperative Nash game, proved to be a potential game.

The paper is interesting and well presented.

However, I have to say that I don't understand how the UAV is taken into account. I try to be clearer: starting from the results, UAVs are stationary points in the space, without dynamics neither kinematics. In fact, in the equations describing the model and the algorithms, the variable position of UAVs is not taken into account, but the clustering algorithm should take into account it (I think).

Can you make clearer how the UAV (as model or object) is taken into account? If your model is static, it should be written, otherwise it needs some statements to explain better and, probably, a result with moving UAVs can be useful to prove the effectiveness of the algorithms.

Furthermore, I'd avoid the first phrase in the abstract: "UAV has been widely used in various networks due to its advantage of high maneuvrability", because the high maneuvrability is not typical of any UAV. If you are speaking of multi-rotors or helicopters, you have to be clearer, because the need of "rotary-wing" UAVs is stated only in the Section 2.

Best regards

Author Response

Dear reviewer,

Feel your valuable guidance. I have modified and adjusted my manuscript based on your comments. The revised manuscript incorporates all the reviewers' comments. For each modification of the original manuscript, we have highlighted it in the revised version, and the line number or position description can be seen in the cover letter. The following is the reply to your comments:

 Point 1: Can you make clearer how the UAV (as model or object) is taken into account? If your model is static, it should be written, otherwise it needs some statements to explain better and, probably, a result with moving UAVs can be useful to prove the effectiveness of the algorithms.

Response 1: In our model, the UAV is considered as a stationary aerial BS and their locations are related to channel calculation and clustering result and finally impact user-rate. In fact, we consider a static-UAV scenario and the UAVs are assumed to be static during the algorithm (we also made this statement in the manuscript, see line 123 ). However, the algorithm is also applied to the UAV-moving scenario, which is not studied in our research and regarded as a special case of our algorithm. Although the UAV is moving, it can be regarded as stationary in a very short time. But we fear that the constant movement of UAVs makes the current clustering result not applicable to the future UAV location. We will study the CoMP technique combining moving-UAV scenario and try to figure out the solution in the next step.

Point 2: I'd avoid the first phrase in the abstract: "UAV has been widely used in various networks due to its advantage of high maneuverability", because the high maneuverability is not typical of any UAV. If you are speaking of multi-rotors or helicopters, you have to be clearer, because the need of "rotary-wing" UAVs is stated only in the Section 2.

Response 2: The first sentence has been revised, please see line 1 of abstract.

Best regards

Reviewer 2 Report

This is a good paper to the result and discussion section. The performance metrics which have been used here, have not been clearly discussed in introduction or lit review (e.g. Cluster Size must have been discussed throughout the paper as I think is one of the most important performance metrics in this paper, another examples user rate which although its connection to ZF precoding has been mentioned but not discussed throughly.) Also, whenever you are reporting average please report standard deviation and discuss about that as well. 

Author Response

Dear reviewer,

Feel your valuable guidance. I have modified and adjusted my manuscript based on your comments. The revised manuscript incorporates all the reviewers' comments. For each modification of the original manuscript, we have highlighted it in the revised version, and the line number or position description can be seen in the cover letter. The following is the reply to your comments:

Point 1: The performance metrics have not been clearly discussed in introduction or lit review (e.g. Cluster Size must have been discussed throughout the paper as I think is one of the most important performance metrics in this paper, another examples user rate which although its connection to ZF precoding has been mentioned but not discussed throughly.)

Response 1: We add the discussion of average cluster size in the abstract(line 12), introduction(line 81, 114) and conclusion(line 279 ). We also add the thorough discussion of user rate performance in line 132 .

Point 2:Also, whenever you are reporting average please report standard deviation and discuss about that as well. 

Response 2: We add and discuss the standard deviation performance of cluster size in Figure 5 and line 248-253.

Best regards

Reviewer 3 Report

This article describes interesting clustering algorithms for a group of UAVs. The article certainly has a scientific novelty and is of considerable interest to readers. However the article is not free from shortcomings.

  1. The authors use the huge figure 1 to illustrate the idea in the article. However, the information value of this figure is small (it is repeated in figure 2). However, in equations 1 and 2 there are also geometric parameters of the model that could be illustrated in figure 1. This will increase the readability of the article
  2. In paragraph 6, the authors draw conclusions about the regularities of the algorithm based on a small number of computational experiments. However, if in one experiment three clusters were formed in one case, and in another two, this does not mean that it will be the same in another experiment! For correct results and conclusions, you need either a larger number of experiments, or a larger grouping of UAVs in the model. And figure 4 is only good for illustrating a typical situation.

Author Response

Dear reviewer,

Feel your valuable guidance. I have modified and adjusted my manuscript based on your comments. The revised manuscript incorporates all the reviewers' comments. For each modification of the original manuscript, we have highlighted it in the revised version, and the line number or position description can be seen in the cover letter. The following is the reply to your comments:

Point 1: The authors use the huge figure 1 to illustrate the idea in the article. However, the information value of this figure is small (it is repeated in figure 2). However, in equations 1 and 2 there are also geometric parameters of the model that could be illustrated in figure 1. This will increase the readability of the article

Response 1: We put additional information in Figure 1 for clearer explanation, including UAV and user's position, and elevation angle.

Point 2: In paragraph 6, the authors draw conclusions about the regularities of the algorithm based on a small number of computational experiments. However, if in one experiment three clusters were formed in one case, and in another two, this does not mean that it will be the same in another experiment! For correct results and conclusions, you need either a larger number of experiments, or a larger grouping of UAVs in the model. And figure 4 is only good for illustrating a typical situation.

Response 2: We have illustrated the original Figure 4 as a typical situation (line 226 ), and we have conducted multiple experiments to obtain the average and variance performance of the number of clusters, please see Figure 5 and line 235-253.

Best regards